# Identifying key features of digital resources used during online science practicals

**Vanda Janštová**[1]*, **Petr Novotný**[1], **Irena Chlebounová**[1], **Fina Guitart**[2], **Ester Forne**[2], **Montserrat Tortosa**[3]

**1** Department of Biology Education, Faculty of Science, Charles University, Prague, Czech Republic, **2** CESIRE Department of Education, Generalitat de Catalunya, Spain, **3** INS Sabadell, Juvenal, Sabadell, Spain

* vanda.janstova@natur.cuni.cz

**Data Availability Statement:** All relevant data are within the paper and its Supporting Information files.

## Abstract

As in our everyday lives, we use digital elements as part of formal and informal education. To serve their educational purpose well, systematic research is desirable to identify and measure their characteristics. This study focuses on science practicals, which are complex and vary in organizational settings and specific arrangements, including usage of digital elements. We describe the digital resources on which the online instruction of science practicals during the COVID-19 forced lockdowns was built, and their key characteristics were identified. Data were collected from science teachers in Slovakia, Czechia, Slovenia, France, and Spain. The teachers shared the web resources they used and that they would recommend, together with a description of the resources. We recorded 89 inputs representing 50 unique resources. Teachers preferred free resources, mostly for knowledge revision, and newly discovered 36% of them due to forced distant teaching. The best evaluated resources were those supporting interaction (especially among peers), focused on teaching subjects and/or ICT, ready to use, and with a clear structure. The resource most frequently mentioned and used in more than half of the countries was PhET (Interactive Simulations for Science and Math) which provides free simulations of scientific principles. Other characteristics mentioned in the literature (e.g., supporting creativity and independent solving, connecting different levels of organization, authenticity, flexibility) were not that important for the overall rating.

## Introduction

As in other sectors of human activity, education in schools is subject to changes associated with the growing inclusion of information and communication technology. For decades, science education has been using digital elements, such as videos and other visual materials or domain-specific simulations, so that by this time there are already digital natives among members of the teaching profession. This development, together with technological advances in connectivity, hardware availability, and the vast array of available resources, has removed the major obstacles that were often considered objective limiting factors to the introduction of

**Funding:** All authors were funded by Erasmus+ project 'My Home - My Science Lab'. The funders had no role in study design, data collection and analysis, decision to publish, or preparation of the manuscript.

**Competing interests:** The authors have declared that no competing interests exist.

technology into the classroom. These resources are now available mostly in an online form; they are easily accessible in school and as homework. Today, they can hardly be seen as a technological novelty or innovative element of teaching, rather we consider them as a natural part of the educational process that corresponds to today's educational reality.

The use of the digital resources (tools that are part of computing environments to support education [1] and are used by teachers) varied in frequency and modality depending on the personality of the teacher, the technological equipment of the school, and also the school philosophy. These factors also influenced the types of digital resources used in education and the specific methods and media. We focus on the practice-oriented components of science teaching, by which we mean hands-on exercises, lab work (referred to as *science practicals* in this study) which, at least in our experience, have been implemented online rather infrequently.

A fundamental change was caused by the COVID-19 pandemic and the transition of schools from conventional to online learning in many developed countries of the world, including in Europe. This change has affected all teachers equally, regardless of their previous attitude towards digital technologies, creating a unique experimental and research space that has attracted the attention of many studies.

With the transition to online teaching, teachers had to modify science practicals to a form suitable for online teaching. The online form was then implemented even by teachers who were not technology experts. Such a necessity caused by external influence dramatically broadened the teachers' experience with the online resources on offer, their possibilities, and ideas about the opportunities for use in their own teaching. As shown by [2], to adopt learning that includes (mobile) computing devices, it is necessary to go through several stages, where two are critical ones: i) introduction and gaining practice, and ii) implementation of the tools into lessons. All of the teachers in the European countries that we focus on have gone through both of these stages while distance teaching, including those who teach effectively, but have so far tended to avoid using technology in their teaching. This provides an opportunity to gain insight into the teachers' evaluation of the digital resources, that is, to answer the question of which digital resource characteristics are markers for teacher acceptance and use in teaching, suppressing the bias of technology acceptance [3, 4].

As noted by [5], the online learning environments were often originally used in hybrid or even in-person settings, which could compensate for their weaknesses and still benefit from the positives, for example, greater flexibility [6]. Therefore, when designing online learning environments, socio-emotional processes have not been the main focus [7]. As a consequence, reduced and impeded social interaction was among the most pressing concerns among students and teachers during the COVID-19 pandemic [8–10].

## Theoretical background

### Digital technology in science classrooms

When the COVID-19 pandemic began, teachers in European countries (and not only them) had to quickly adapt their courses from face-to-face to online ones due to massive lockdowns. It was a great challenge for them but the transition to technologies in education was driven by need and affected all teachers non-selectively [11]. Because they had to react nearly on a day-to-day basis, teachers tended to focus on gaining the general technological skills specific for online instruction followed by content (re)transformation and paid less attention to the integration of technology with pedagogy and content [12]. This integration is important for effective teaching and, in general, similarly to other teaching settings, covers and combines cognitive, affective, behavioral, and social aspects through communication and collaboration to engage the learner [13]. The theoretical ground for educational technology and consideration of its interaction

with other described components in the form of the concept of technological pedagogical content knowledge (TPACK), [14] was available before COVID-19. This framework has been used in many of the studies investigating teachers' use of technology and is probably the most commonly-used theoretical framework in the context of both research on technology in schools and the professional development of science teachers [15]. Its weaknesses are fairly well known in the context of science and biology and are summarized, for example, by [16]. We assume that they can also be applied to other disciplines, among whose lessons, for example, Walan found no difference [17]. A very important aspect when studying technologies using the TPACK framework is to take into account the dynamic nature of the domain–many technologies (pencil, book) are used routinely, hence sensu Cox and Graham [18] we understand them as transparent technologies. The extension of the TPACK framework over the extension of Shulman's work [19] on pedagogical content knowledge (PCK) lies only in emerging technologies, those that are being investigated or introduced into a learning environment. However, in our view, the distinction between transparent and emerging technology has shifted significantly since the establishment of the research field. Most of the technologies whose uses and characteristics researchers observe are difficult to consider as emerging technologies, be they email, online resources, or collaborative cloud space.

These technologies are a routine part of the administrative side of a teacher's job, and to view them as emerging in teaching (let alone after the COVID-19 pandemic during which technology was a de facto core part of teaching) seems to us to be a freezing of discourse (e.g., the use of TPACK as a theoretical framework for analyzing the use of PhET [20] or adding verbs typical for the digital environment (like programming, animating, podcasting, tagging, testing, advanced searching) to Bloom's taxonomy [21]). There has been a fundamental transformation whereby the use of technology in school is no longer "driven more by the appropriateness of technology rather than the demand of pedagogy and teaching of subject matter" [22], but attitudes toward technology are predictors of how technology is used, not whether it is used. Thus, we are in a situation where the use of technology is an everyday practice and separating research into situations with/without (modern) technology gradually loses its widespread importance. Even for the technologies that are closest to be called emerging, i.e., probably AR/VR, it can be documented that they have been used in school settings for more than a decade [22] and the crucial point ceases to be the technology itself. Therefore, what is important for today's schools is not whether they use technology and which ones, but, in the words of [23] "convert artifacts [e.g. digital resources, that is, an entity or product of human creation, with embedded knowledge] into epistemic tools". The cited work provides a fundamental theoretical framework for the design of digital resources culminating in three guiding principles that relate to both the resource itself (guiding principle 1 –"A digital resource needs to have other aggregated epistemic artifact(s) to be used for educational purposes"–that includes some kind of "exploration guide", methodological hints for teachers), to the work of the teacher (guiding principle 3 –"A digital resource must be inserted in an orchestrated chain of artifacts, used in a setting of learning in the context of epistemic practices") and to both at the same time (guiding principle 2 –la "A digital resource becomes a tool if used effectively to solve a task/problem in a setting of learning context"). Other similar frameworks of thought, which in our opinion are much better suited to the needs of studying technology in learning than, e.g., the TPACK framework, have been developed based on the experience of pandemic capping. In our work, we have used a framework formulated by [24] focused on online arrangement. We consider this framework to be close in idea to our subject of interest, online science practicals. It describes the foundation components of online pedagogy, namely five pillars that grow out of the principles of learner-centeredness [25], constructivism [26], and situated learning [27]. According to [24], these five pillars include i) *cultivating relationships*

understood as interactions that build relationships both within school (among students and between students and teachers) and outside school (within the family, between students, and nature, students and society); ii) *active learning* e.g., choice in the path to the learning goal, independent solving; iii) *learner autonomy* and supporting creativity where learners are seen as co-creators who seek information and build connections; iv) *personalization* which values, respects, and accommodates learner differences by providing learning materials in a variety of formats and modalities and allows independent exploration, time, and place flexibility; v) *mastery learning*, which is described as a customized curriculum based on assessment data that allows learners to see the progress they are making and gives them constructive and consistent feedback.

## Science practicals

As in [28], we understand science practicals as an overarching term including any type of activity linked to science education in which students (individually or in groups) manipulate and/or observe real objects no matter if they use a "cookbook", inquiry, or other approach. They can be held in laboratories (e.g., flame tests, microscopy), outdoors (e.g., plant identification), or in any other environment. Science practicals have had an important role in science education and have been shown to be a challenging method even in face-to-face settings, as they can easily turn out to be ineffective [29]. Their implementation is often considered to be more demanding even in face-to-face teaching: preparation, safety standards, organization, interpretation of unexpected results, and unpredictability. They are effective in making students manipulate objects according to a recipe [30, 31], typically offering a unique opportunity of working in a setting of smaller class sizes, higher instructor-to-student ratios, and the diverse learning possibilities laboratory classrooms offer. Practicals can be effective and develop conceptual understanding when it is "hands on" and "minds on", and when teachers explicitly guide students to link these two essential components of practical work [32, 33]. Also, in addition to improving knowledge, practical work has been shown to improve attitudes toward the subject [34, 35], interestingly in all of the following settings: traditional (not using computers), computer-supported laboratory, and computer simulation [34].

The interest of researchers in science practicals settings led to the formulation of the design principles of student resources. These principles were developed regardless of forced online teaching by [36] and include the ones relevant for our study which we highlight: i) *safety*–training to work in safe settings, in the online environment understood as the possibility to safely study dangerous or inaccessible features/processes; ii) *authenticity*–being based on real word/research situations; iii) *flexibility*–provides a variety of topics and difficulty levels.

To map the emerging situation, [37] analyzed different types of adjustments made to laboratory curricula and activity types due to the COVID-19 pandemic (e.g., different types of experiments, simulation, pre-collected data analysis, planning, reviews, etc.) and the reported immediate effect on students. The lack of hands-on laboratory experience was reported to be counterproductive to some types of learning and participation, such as understanding procedures, analyzing nonideal data, evaluating the trustworthiness of the data, troubleshooting, making evidence-based conclusions, proposing experiments, or predicting results [37, 38]. However, some goals could be achieved remotely [37] with the recommended setting, for example, including peers in the learning formulated by [24, 36]. Specific recommendations for science practicals were introduced by [37–39] and are in agreement with the general rules. [40] point out that the fact that most students today can use their smartphones for data recording, taking pictures, and connecting to online apps such as virtual simulations helped to reach the science practicals goals.

## Activity types in online science practicals

In addition to a description of the features of a digital resource and its potential to meet the needs of online learning, it is also useful to look at the form in which the resource is implemented in teaching, i.e., in what type of technological enhanced activity it is incorporated. For the typology of activities, we used [41] as a basis, selected those that are typical/characteristic for science practicals, and modified them slightly when confronted with more up-to-date resources. We especially considered that the digital resources used for face-to-face labs had to be altered (or even replaced by a new format) from the basis for distance teaching, and instructions began to emerge shortly after the first COVID-19 forced lockdown. This modification can perhaps be seen as a mere change of modality [42] or as a new type of activity given that it is a paradigmatically new teaching approach [41]. The difference between online and face-to-face teaching is not simply a change of communication channel, but a qualitatively different type of work. The role of the digital elements in the online setting can be different from face-to-face settings and take on new functional roles, where, for example, a quiz is not only an activation and feedback tool but also a quick way of presenting the results of measurements or an assumption of the outcome of the intended observation. Therefore, we are closer to viewing it with respect to the total givenness of the modality as a new type of activity, but consideration of this issue is beyond the scope of this paper.

We have identified these relatively specific activity types for online science practicals: I) *Investigation*, experiments, inquiry, e.g., in kitchen settings which can be combined with online tools, for biology possibly including individual nature visits, e.g., sample collections outdoors like making a (digital) herbarium, includes remote experiments [37, 41, 43]. II) *Virtual reality*, which provides realistic interactions with 3D computer generated learning environments, or digital lab environments which are entirely in a virtual interface (e.g., Labster, Beyond Labz). Virtual laboratories can provide a safe learning environment accessible online, helping to better understand the experiment as a whole. The recommended setting is to use virtual laboratories before the real ones as a preparation for face-to-face laboratory experiments, as they lower stress level and anxiety and improve learning [44–47]. And this is where the paradigm shift became apparent, where their role was seen as an adjunct to the face-to-face setting, but during forced online learning it became a separate form. During distant teaching, virtual laboratories were often substitutes for wet labs. III) *Computer simulations* [41] have been shown to improve conceptual understanding of chemistry [48] and physics [49]. The second study shows an example where students using computer simulations explicitly modeling electron flow understood the electricity concepts better and also could assemble the real circuit and describe how it worked better than students working in a real laboratory. IV) *Data analysis and computational science* (using computers to solve science problems) [41] demonstrate the current approach in all science disciplines–solving, for example, molecule structure, molecular genetics [50] or enzyme kinetics [51]. V) *Video-based learning* can be in many styles ("images" according to [41]), from simple pre-recorded video experiments to real-time lab recordings [52]. VI) *knowledge revision*, like answering questions, taking a test or a quiz [41] have been shown to significantly improve knowledge retention [53, 54]; VII) *games* including a story line and rewards when students have to reach a goal either individually or in a team [55]; VIII) *modeling and construction* when students learn by doing, e.g., construction a measurement tool with Arduino followed by use of the mobile app "Arduino Science Journal" [56] or building a model [41]; and IX) *virtual field* trip when the "visitors" do not go to the actual site but "visit" it through a virtual tour [41, 57, 58].

New digital resources are continuously emerging and derived from new activity types, worthy of the interest of the general teaching public. Forced online learning has accelerated the

"coevolution" of digital resource developers and their users, clarifying the requirements for their functions and teachers' demands for the inclusion of aspects (e.g., communication) that might have been neglected before. Therefore, we consider it important to be able to define their qualitative characteristics and to know which ones are decisive when teachers decide whether to adopt them in their future (face-to-face) practice. We start from the principles formulated by [24, 36] together with activities [37, 41] and combine them to collect data about digital resources used during online science practicals.

### Aims

Our aim was to identify digital resources that were adopted by science teachers during lockdown and incorporated into online science practicals; and to propose an interpretation of their shared characteristics which made them useful to teachers when planning effective online science education.

Specifically, we wanted to find answers to the following research questions that describe the key features of digital resources used during online science practicals:

1. What were the most important characteristics of the digital resources that led teachers to use them?

2. What types of activities were enhanced by using technologies?

3. What characteristics of the resources make the difference in the appropriateness of the individual activity type?

## Methods

We used an online survey to collect data from secondary school science teachers.

### Sample and sampling

Digital resources were collected in 5 EU countries: Czechia, Slovakia, Slovenia, France, and Spain from in-service science teachers using an online survey (GoogleForm). The teachers were contacted based on their contacts and possibly also previous cooperation, and asked to share only activities they knew, had used, and considered useful for online science teaching. As has been shown [59] even similar low stakes, e.g., via email provides relevant results. As we aimed at describing the digital resources which teachers used, we omitted the idea of asking for the resources they had tried and rejected. We decided so also because one can hardly assume that teachers, when searching for resources, will remember which resources they have not accepted, let alone attach relevant assessments to them retrospectively [60].

### Instrument

Teachers provided the name of the resource, hyperlink, overall rating of the resource (scale 1–5, the more points, the better rating), information about modality (synchronous, asynchronous, both) and the subject (biology, chemistry, physics, geology, geography). The appropriate educational levels ISCED 1–3 were recorded together with information about availability (if it was free or needed payment).

The pedagogical characteristics of the resources were sampled in the following sections: 1) type of *activity* (experiment, video, knowledge revision, augmented reality, simulations, etc.)

with classification based on literature, see Table 1; 2) *pedagogical suitability*; 3) *interactions*; and 4) specific *added value* which are described below.

We provide a short definition, and the literature from which we draw.

In the next section of the survey, teachers reported the *pedagogical suitability* of the resource as the level of their agreement with statements describing the ease of use ("The structure is clear", "The visual is appealing"), suitability of the online tool for teaching science ("This resource helps to teach the subject") and teaching ICT ("It helps to teach an important ICT skill') and other skills ("It helps to teach something else important (except subject topic and ICT)"), and willingness to use the activity based on this resource as is or after modification during online teaching or standard teaching ("I would use this resource as it is, even in standard teaching mode", "I would use this resource as inspiration and modify it for standard teaching mode") on a 5-point Likert scale. Teachers rated the support for *interactions* (cultivating relationships, [24]) among students, students and teachers, students and society, students and families, students and nature, to assess the support in building relationships and community (using a 5-point Likert scale). Practicals are appreciated for specific *added values* as shown by the research; the extracted categories used for this study are summarized in Table 2. Except for the categories based on principles formulated mostly by [24, 36], we reported two more: providing worksheets and connecting different organizational levels. Providing worksheets to students is often part of science practicals [64], and ready-to-use worksheets are typically appreciated by teachers. Connecting different organizational levels is necessary to understand science and is described as an issue especially in biology [65, 66].

The authors also additionally recorded whether the resource was aimed at higher levels of cognitive processes of the revised Bloom taxonomy in factual and/or conceptual knowledge dimensions [21, 67] or at "remembering and reproducing". The simultaneous assignment of the resource to the knowledge revision and Bloom's "remembering and reproducing" categories was taken as a coding check.

## Data cleaning

The data was cleaned in Excel. The activity types which were not mentioned or were mentioned only once (*A_virtual trip*, *A_modelling*) were not included in the analysis. If the question allowed multiple answers, the format was transformed to dummy variables. All the answers were translated into English. Inconsistent terminology used by teachers was unified.

## Statistical analysis

R v.4.1.2 [68] was used for further analyses of data collected. For the description of *pedagogical suitability*, we proposed two items–I_17+I_18 (I_17 "I would use the resource as it is, even in standard teaching mode", I_18 "I would use this activity as inspiration and modify it for standard teaching mode') which we considered as complementary, i.e., negative correlation. We expected a split between resources that are ready-to-use and resources that need more or less modification to be accepted by teachers as suitable. We tested this assumption with the Spearman correlation coefficient.

Principal component analysis (PCA) computed on the correlation matrix was used to describe the variability of the *pedagogical suitability* component. Data were not transformed as all items describing *pedagogical suitability* were collected in the same manner as a 5-Point Likert scale [69]. The decision on principal component count to be analyzed was based on a scree plot of the explained variance of each of the principal components compared to the amount of variance each variable would contribute if all contributed the same amount.

**Table 1. Activity types used in the survey.**

| Activity type | Definition | Literature | Acronym used in text A_learning digital element |
|---|---|---|---|
| **Experiment including remote experiment and demonstration** | Students explore the topic, control variables, and have to observe and interpret the results | [37, 41] | A_experiment |
| **Knowledge revision** | Pre-prepared knowledge revision in a different form such as answering questions, taking a test or a quiz | [41, 53] | A_revision |
| **Literature/data-based analysis** | Drawing conclusions from the literature, analyzing pre-collected sample data | [37, 38, 41] | A_literature |
| **Virtual field trip** | Virtual tour/site visit | [41, 57, 58] | A_virtual trip |
| **Video-based learning** | Simple video recordings, narrative/voice-over lab recordings, and real-time delivery | [37, 38, 41] | A_video |
| **Simulation** | Simulations of processes focused on teaching of the concepts | [37, 41, 61, 62] | A_simulation |
| **Virtual/augmented reality** | Augmented reality (AR) and Virtual Reality (VR) providing realistic interactions with 3D computer generated learning environments | [38, 45, 63] | A_virtual reality |
| **Modeling, constructing** | Learning by doing, e.g., devices build with Arduino, building a model | [41, 56] | A_modelling |
| **Games** | Reaching a goal with an identity created for the game, can be played in teams | [55] | A_game |
| **Investigation** | Inquiry; students investigate a scientific phenomenon, different levels (open, structured, individual, in groups) | [36, 41] | A_investigation |

*Interaction support* provided by the resource was analyzed using a divisive hierarchical clustering. For clustering, we used the cluster package [70] with the similarity coefficient metric [71], which is suitable for categorical data [72]; evaluation of the number of interpretable clusters was based on a scree plot of internal cluster variability and a silhouette index comparison [73].

A linear regression model to predict the overall score from individual subscales was fitted. We use a backward elimination to choose the final model.

To investigate the effect of subcategories from the *added value* group, we used the non-parametric Kruskal-Wallis test with Bonferroni correction on multiple testing.

All statistical tests were performed at the chosen significance level of 95%. Effect sizes were labeled following recommendations [74]. 95% Confidence Intervals (CIs) and p-values were computed using a Wald t-distribution approximation.

To analyze the relationship between features of the digital resource and activity type (in which the resource was used by the teachers), we used multi-label classification methods by package *mlr* using conditional random forests as a learner. The training set consisted of 50

**Table 2. Resources *added value* overview.**

| Added value | Definition | Literature |
|---|---|---|
| **Supporting independent solving (active learning)** | Choice in the path to learning, active problem-solving opportunities | [24] |
| **Supporting creativity (learner autonomy)** | Supports coming up with novel solutions, learners have autonomy and are viewed as co-creators | [24] |
| **Differentiation of difficulty** | Provides learning materials at a variety of difficulty levels | Part of personalization [24] |
| **Safety** | Allows the exploration of otherwise dangerous/inaccessible processes | [3, 36] |
| **Authenticity** | Including features from real situations/research | [36, 37] |
| **Flexibility of topic** | Providing a variety of options, resources that can be readily reconfigured; covers multiple disciplines. Part of the personalized learning process | Part of personalization [24, 37] |
| **Providing worksheets** | Ready-to-use worksheets provided | [64] |
| **Connecting different organizational levels** | Exploring/connecting more organizational levels (e.g., population–organism–organ–tissue–cells–organelles–molecules) | [65, 66] |

randomly selected resources, while the remaining 39 served as the test set. All of the descriptive features that we had available were used as features, in addition to the overall teacher ratings. The metric of performance used was Hamming Loss, that is, a proportion of activity types that are predicted incorrectly, following the definition by [75].

## Results

We collected a total of 89 submissions representing 50 unique resources (see S1 Appendix for data). A significant number of the resources were in national languages with a local impact. Only two resources were mentioned more than three times. The resource most frequently mentioned (16 times) and the only one mentioned by teachers in at least three of the five countries was PhET (https://phet.colorado.edu). Half (49.55%) of the resources were mentioned only once. Teachers reported using nearly half of the resources even before the pandemic (48%), 36% of them were newly discovered due to forced online teaching, and the remaining 16% were resources which teachers knew but did not use before the pandemic. Most of the resources reported (82%) are for free, 8% are available only after purchasing, or there are major differences between the free and paid versions, and 10% of the resources have minor differences between the paid and free versions. Most (76) of the resources implicitly had science content (e.g., DR_25 PhET, DR_5 periodic table of elements), 13 of the resources could have any content depending on the teacher (e.g., DR_88 Mentimetr, DR_29 Kahoot, DR_85 Padlet).

### Activity types

The teachers were asked to state an activity type for which they use the resource. Most frequently, it was *A_revision*, followed by *A_simulation (e.g., population dynamics; input data, factors, etc., see results)* (Table 3). More than one activity could be chosen and 3 was the most frequently used number of them in their answers. The highest number of activities stated for one resource was 6.

In the situation where multiple answers were chosen at the same time, the combinations of *A_revision + A_simulation* or *A_revision + A_investigation* were the most common (N = 24); an overview of the combinations that occurred in more than 15 cases is given in Table 4.

### Pedagogical suitability

The highest sampled score of *pedagogical suitability* was 35 (out of 35 possible), and the lowest *pedagogical suitability* score was 18 with the median value 27. The three most highly valued resources were all targeting the PhET project (DR_25, DR_70, DR_71).

**Table 3. List of activity types with the frequency among resources.**

| Activity type | Count [%] |
|---|---|
| A_revision | 60 [67.4%] |
| A_simulation | 40 [44.9%] |
| A_investigation | 29 [32.6%] |
| A_literature | 20 [22.5%] |
| A_video | 20 [22.5%] |
| A_experiment | 15 [16.9%] |
| A_games | 11 [12.4%] |
| A_virtual reality | 8 [9.0%] |

The percentage is counted from the number of resources (n = 89).

The two survey items–I_17+I_18 (I_17 "I would use the resource as it is, even in standard teaching mode", I_18 "I would use this resource as inspiration and modify it for standard teaching mode")–were designed to be complementary but turned out to be independent. The Spearman's rank correlation rho between I_17 and I_18 is positive, statistically not significant, and very small (rho = 0.09, S = 0, p = 0.398).

The analysis of the primary components of *pedagogical suitability* showed the primary component PC1 explaining 37.8% of variability, the following axes 15.6% and 14.6% of variability, respectively, see Figs 1 and 2. Therefore, we interpret only the first component, graphically shown in Fig 2. PC1 is loaded with items I_17, I_19, I_20, I_21. That is "use as is", "teach subject", "structure clear", "important ICT skill", see Table 5 for PCA loadings and item wording.

Thus, the features forming PC1 are the main source of variability in the teachers' evaluation of resources (as overall ranking) in a linear model. The model's intercept, corresponding to PC1 = 0, is at 4.02 (95% CI [3.85, 4.19], t(87) = 47.54, p < .001). Within this model, the effect of PC1 is statistically significant and positive (Std. beta = 0.61, 95% CI [0.44, 0.78], t(87) = 7.15, p < .001).

The first primary component explains 38% of variance. The red horizontal line indicates the amount of variance each variable would contribute if all contributed the same amount (14.3%).

The length of the arrows corresponds to the loadings of each item, the colored ellipses highlight the distribution of the resources with respect to their overall teacher rating. The trend to increase the overall rating of the resources with respect to the positive direction of PC1 is clearly visible.

## Interaction support

The highest sampled score of the *interaction support* was 25 (out of 25 possible), and reached by three resources (DR_42 Química en context, DR_70 PhET and DR_71 PhET), The lowest pedagogical suitability score was 5, the median value was 17.

The *interaction support* category formed two distinctive clusters. The chosen number of clusters is supported by the *Silhouette* index, showing that two clusters are most dissimilar in between. The *Elbow* plot does not indicate a clear cut-off and therefore we interpret two clusters (Fig 3). Group A was characterized by interactions that focused inside the school, student-teacher (I_24) and student-student (I_25) interactions which were supported by resources like DR_88 Mentimetr, DR_29 Kahoot, DR_85 Padlet with which activities are typically used synchronously. Group B did not have the school interaction and contained the student's interaction with society (I_26), nature (I_27), and/or family (I_28). Examples can be resources DR_42 "Química en context", or DR_17 "Climate facts" which collects data on climate and climate change provided by scientific institutions (e.g., NASA, Eurostat, national data about weather, and others) and compiles them into graphs and infographics for further use.

**Table 4. Most common pairs of activity types reported together for a single web resource.**

| Learning components couple | Count [%] |
|---|---|
| A_revision + A_simulation | 26 [29.2%] |
| A_revision + A_investigation | 24 [27.0%] |
| A_simulation + A_investigation | 21 [23.6%] |
| A_revision + A_video | 16 [18%] |

The percentage is counted from the number of resources (n = 89).

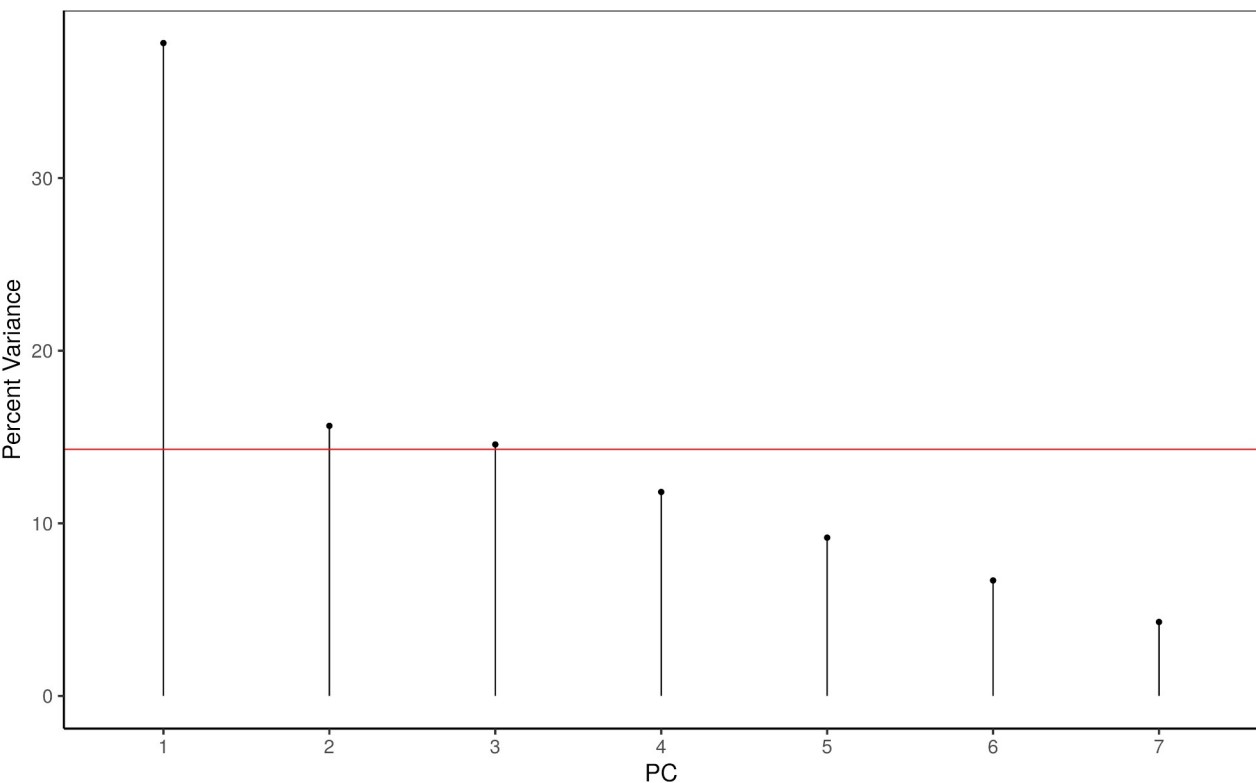

**Fig 1. Explained variance percentage of primary components of the DR *pedagogical suitability* items.**

The peer interaction (I_25) was the most frequently mentioned (57 resources, median = 4) followed by the student-teacher interaction (I_24, 47 resources, median = 4), student-society interaction (I_26, 41 resources, median = 3) together with the student-nature interaction (I_27, 40 resources, median = 3). The interaction mentioned least frequently was the student-family one (I_28, 17 resources, median = 3).

## Added value

The categories of resource *added value* occurred with comparable frequency–I_35 (providing worksheets) was the least represented at 19 resources (21%), and I_29 (supporting independent solving) and I_32 (safety) were the most frequently assigned–these were listed for 45 resources (50.6%), with a median of 37.5.

More than a third, 32 resources (36%), had no *added value* assigned. All categories were assigned to 11 resources (12.4%). The remaining 46 resources (51.7%) had between one and seven categories assigned (Fig 4).

## Bloom's taxonomy

Most (48, 53.7%) of the resources were rated as "remembering" and aimed at a low cognitive level.

## Overall rating

As part of resource reporting, the teachers gave an overall rating, a "number of points" to each resource (the higher the better). The rating was mostly high (maximum 5 points out of 5,

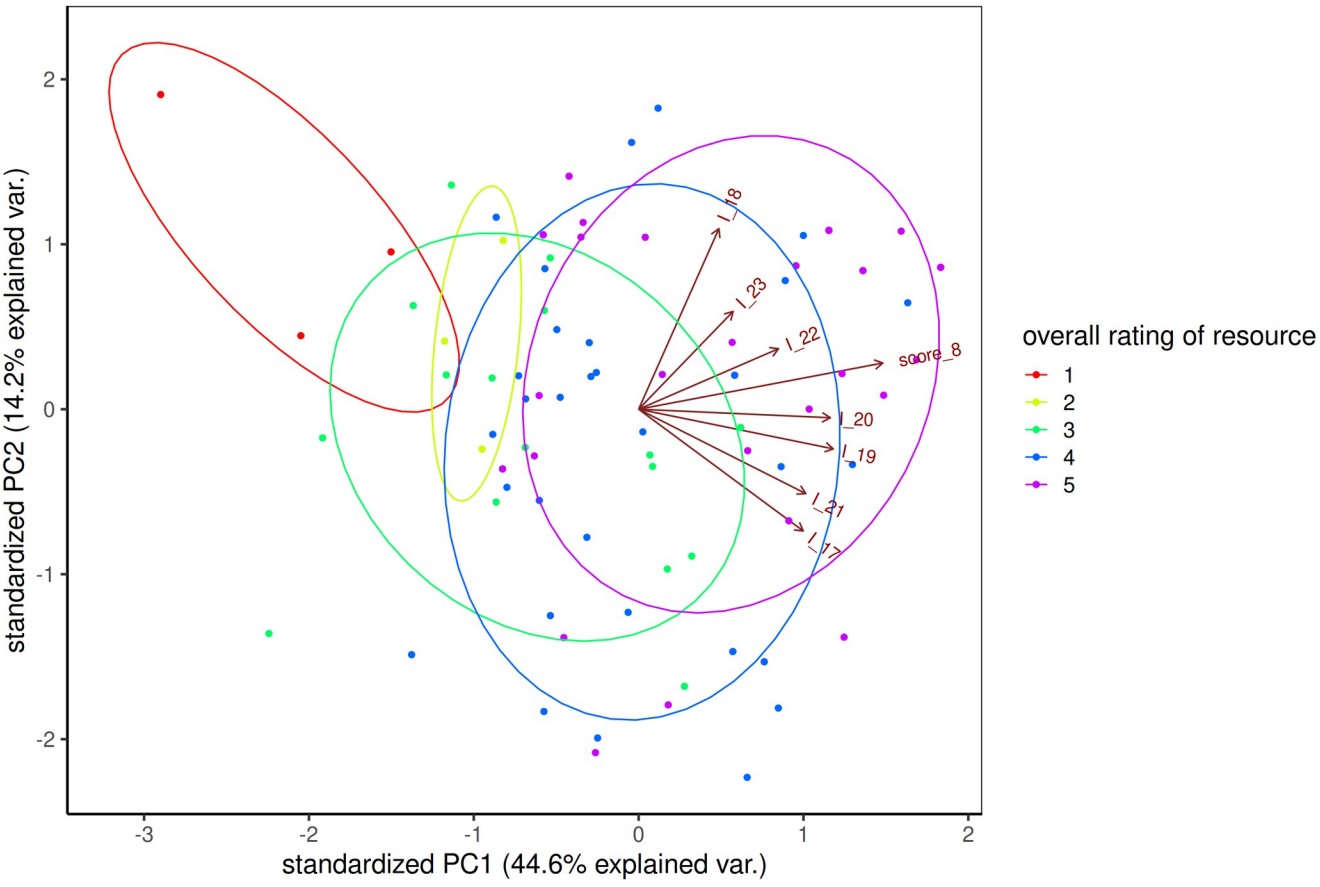

**Fig 2. Visualization of the first and second primary components of the DR pedagogical suitability items.**

median 4 points), as the teachers were asked only for examples of good practice, the resources they wanted to recommend to others.

## Relationship between overall rating of resource and individual subscales

We fitted a linear model to predict the overall rating of the resource. As explanatory variables, we used *pedagogical suitability*, *interaction support*, *added value*, and *Bloom's taxonomy*,

**Table 5. PCA loadings in the first principal component of the DR pedagogical suitability items.**

| Item number | PC1 | Item wording |
|---|---|---|
| **I_17** | **0.4278753** | I would use this resource as it is, even in standard teaching mode |
| I_18 | 0.1373113 | I would use this resource as inspiration and modify it for standard teaching mode |
| **I_19** | **0.5038127** | This resource helps to teach the subject |
| **I_20** | **0.4891261** | The structure is clear |
| **I_21** | **0.4074133** | It helps to teach an important ICT skill |
| I_22 | 0.3062552 | It helps to teach something else important (except subject topic and ICT) |
| I_23 | 0.2126473 | The visual is appealing |

Loadings higher than 0.38 are highlighted in bold, as a theoretical cutoff if all variables contributed equally to that principal component.

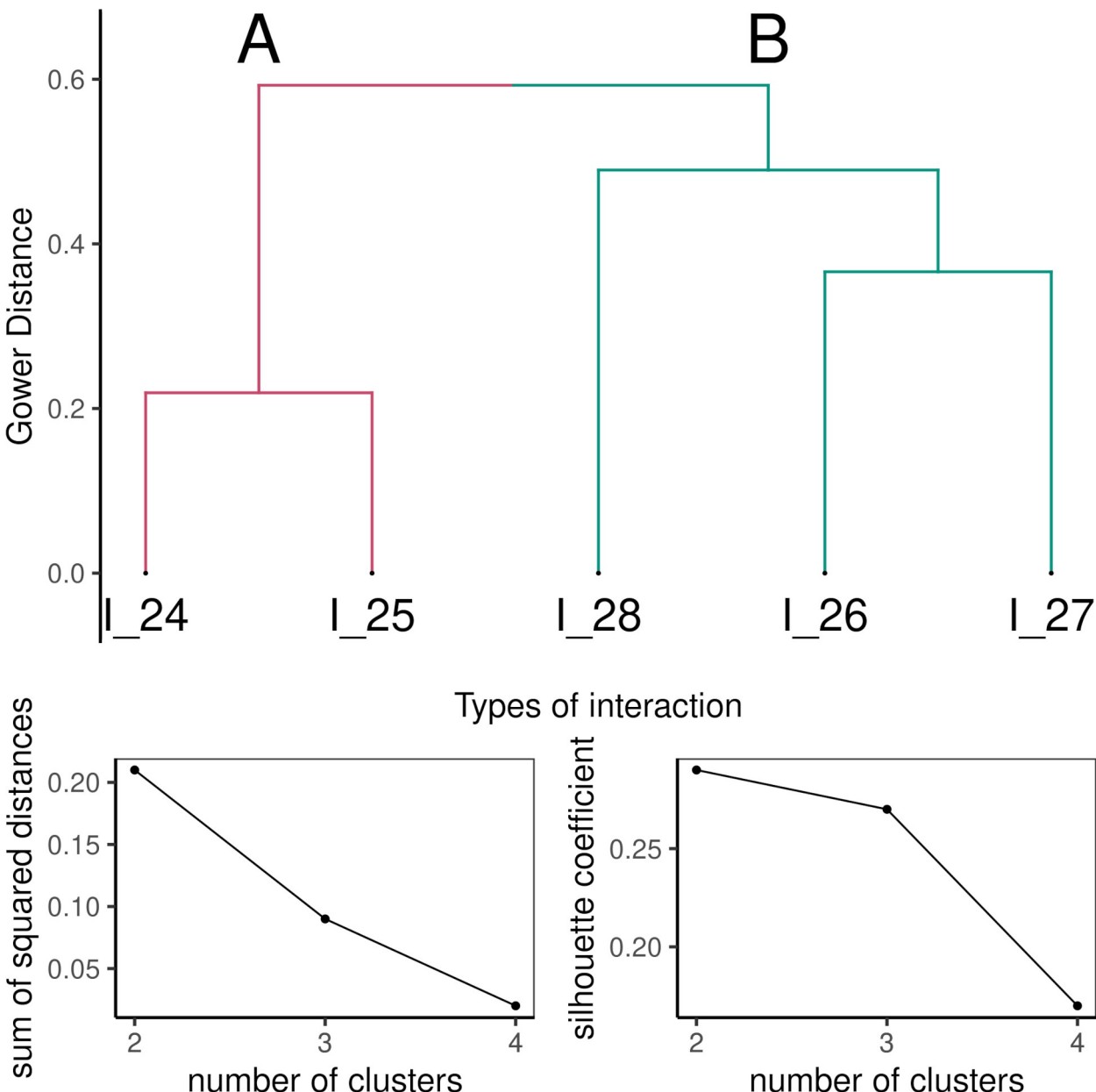

**Fig 3. Interaction support of resources.** First row: Dendrogram of interaction support of resources. The left cluster (marked A) represents interactions in the school context; the right cluster (marked B) represents interactions in the out-of-school context. Second row: Analysis of the elbow scree plot and the silhouette index to select the number of interpreted clusters. The plot on the left represents the internal variability of clusters without sharp breaks. The right plot represents the Silhouette index, the greater the dissimilarity of the clusters, the more dissimilar. The highest dissimilarity occurs only at two clusters.

selecting the model by a backward selection. The accepted model formula is *overall rating ~ pedagogical suitability + interaction support*. This model explains a statistically significant and substantial proportion of variance ($R^2 = 0.38$, $F(2, 86) = 25.87$, $p < .001$). Within this model:

- The effect of *pedagogical suitability* is statistically significant and positive (Std. beta = 0.44, 95% CI [0.22, 0.66], $t(86) = 4.02$, $p < .001$)

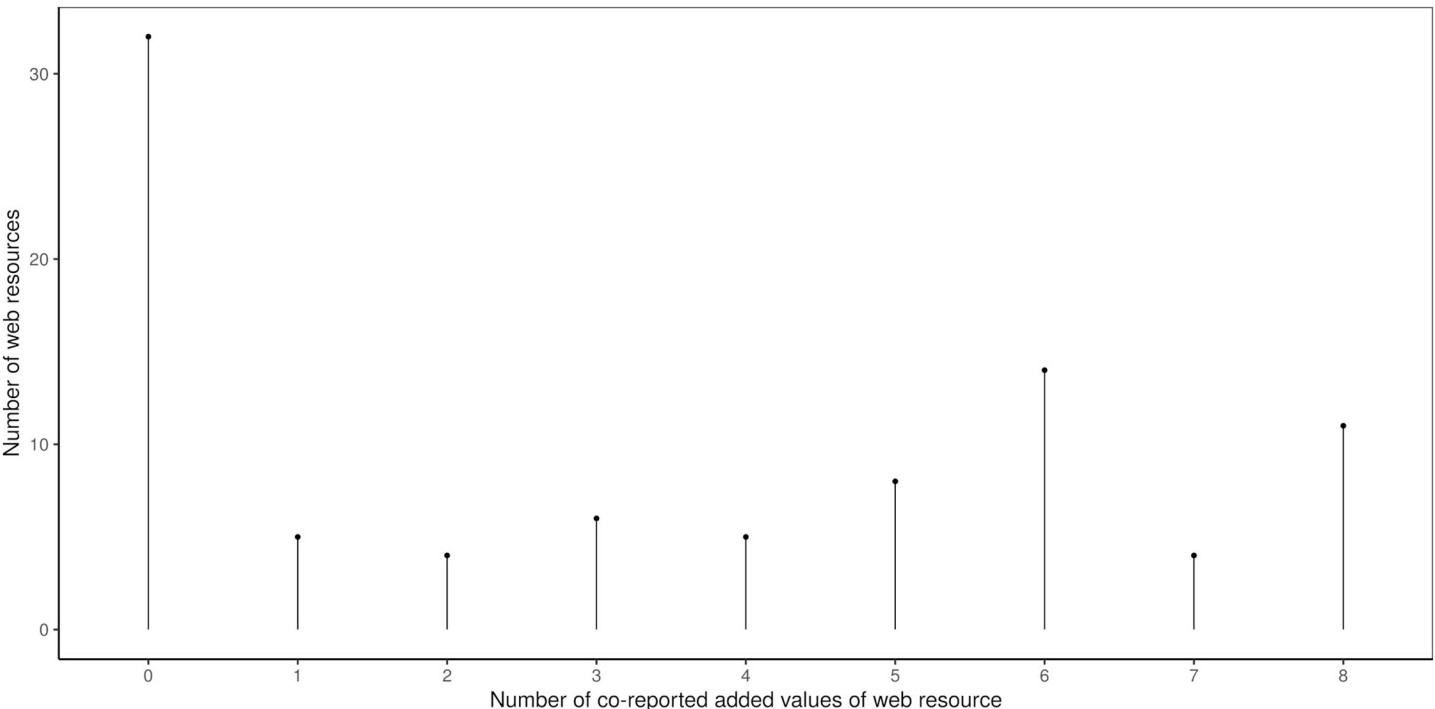

**Fig 4. Number of added value categories listed together.** More than a third of the resources had no added value assigned; see the first bar; otherwise there was considerable variance in the number of *added value* categories listed together.

- The effect of *interaction support* is statistically significant and positive (Std. beta = 0.23, 95% CI [0.02, 0.45], t(86) = 2.14, p = 0.035)

## Relationship between features of digital resource and activity type

The multi-label classification model reached the performance measured by HammingLoss = 0.225, that is, more than 75% of resources were assigned to the correct activity type. This is a good success rate for the model in a situation where we assume considerable variability in the data. The success rate of the single activity classification varied. Virtual activities and augmented reality were the easiest to predict, whereas knowledge revision or simulation was

**Table 6. Multilabel classification performance measure per activity type.**

| Activity type | Accuracy |
|---|---|
| A_games | 0.950 |
| A_virtual reality | 0.900 |
| A_quiz | 0.850 |
| A_literature | 0.825 |
| A_video | 0.775 |
| A_experiment | 0.700 |
| A_investigation | 0.700 |
| A_revision | 0.650 |
| A_simulation | 0.650 |

the most difficult, see Table 6. Among the entire set of available features, the most important features used by the classifier were (sorted descending) I_36 (connecting different organizational levels) = 31, I_37 = 37 (Bloom's taxonomy), I_X35 = 30 (providing worksheets), I_30 = 25 (supporting creativity), I_25 = 10 (student-student interaction), I_24 = 10 (teacher-student interaction).

## Description of the most frequently mentioned resources

*PhET* (phet.colorado.edu/) offers free interactive simulations for science and math that can be used in a variety of settings, from demonstration experiments to individual exploration. Students test their hypotheses and answer questions by changing the setting of different variables and observing the results, and gain a deeper understanding of basic concepts. The simulations convey an understanding of very small, dimensional, or fast processes that cannot be easily observed directly. The topics are, for example, neuron function, gene expression, color vision, natural selection, energy forms and changes, pH scale, diffusion, molecules and their polarity or shapes, atomic interactions or balancing chemical equations, the solar system, electrical circuits, the force of gravity, density, vector addition, Coulomb's law, curve fitting, fractions, graphing quadratics, projectile motion, and many more. Registered users have access to teaching materials. The design of the simulations has been tested and proven effective [76–79]. The resource was mentioned 16 times in our survey, typically gained an overall rating of 5 and a positive rating in *pedagogical suitability*. The evaluation of *interaction support* varied among the individual simulations evaluated as different teachers described different simulations or the resource as a whole.

*Kahoot* (https://kahoot.com/) is suitable for those who want to create their own knowledge tests in a fun way. Upon creating an account, the user can be a teacher, a student, or groups of students. Kahoot can be used as a class competition when everyone gets the link at once and also as a challenge for home repetition of the subject content. In this case, Kahoot is open for a limited time (one week) for everybody who gets the link. The students see their own position in the competition at the end of the game. The teacher has an overview of the difficulty of each question and also about the placement of each player. There are more types of questions in the paid version than in the free version. The resource was mentioned 4 times in our survey, typically gained an overall rating of 5 or 4, and would be used even in standard teaching mode; the visual was rated as appealing. The support of the *interaction* within school was mostly rated as primarily positive contrary to the outside interactions. Kahoot represents an example of a web resource which can have any content, not only a scientific one.

*Mozaweb (*https://www.mozaweb.com) is another paid application designed for science and humanities subjects. The registered teacher can use the digital teacher books, presentations, e-learning, 3D animations, videos, and tasks to test the knowledge of the students. The resource was mentioned 3 times in our survey (overall ratings 5, 5, 1), and would be used even in standard teaching mode; the visual was rated as appealing. The *interaction support* was rated as neutral.

*"Chemistry in context"* (Química en context, on Google sites) incorporates competency work in high school to support meaningful learning and motivate students toward chemistry through connections with everyday life. Among the activities, there are many inquiry-experimental ones, some of which take advantage of the potential of data recording and analysis equipment. In general, the activities focus on modeling, inquiry, and argumentation [80]. The resource was mentioned 3 times in our survey (overall ratings 5, 5, 4), and would be used even in standard teaching mode; the structure was rated as clear. The *interaction support* within the school was rated as neutral and outside school (with nature, family and society) as positive.

*"We know the facts"* (https://www.umimefakta.cz) is a tool suitable to repeat or test knowledge from almost all subjects. The school has to pay for the application to have the pupils in a class together and to allow the teacher to give them tasks and see the result. At the same time, individual students can use these activities individually for free; there is only a limit of 77 questions (approximately 30 minutes of revision) per day without registration. The tasks are done in a funny way at different difficulty levels. If teachers want to use the free version of the application, they can ask the students to send them a screenshot of the results. The resource was mentioned 2 times in our survey (overall ratings 4 and 3), and would be used even in standard teaching mode. The *interaction support* within the school was rated as neutral or nonexistent.

The following two apps were mentioned only once, but we consider them worth sharing.

The *geocaching* app (https://www.geocaching.com/sites/adventure-lab/en/) leads users outside. With the help of an application map, everyone sees the places that are near the position where they are standing. Students (without the teacher) follow the path given by the teacher and collect the specified information about the places they visit. There are many prepared places with content about their history or some interesting facts related to them, even in the free version. In the paid version, teachers can prepare their own places with the content they need. The resource can be used as it is, or modified for standard teaching mode. The *interaction support* among peers and with nature was rated as positive.

The virtual *microscope* app (https://www.ncbionetwork.org/iet/microscope/) is designed to familiarize students with the microscope in advance to using one. Everybody with the link gets into the application, for free, to see the structure of the microscope and can explore selected plant, animal, human, and bacteria microscopic slides. The slides are also described, and the user can change the magnification, adjust the light, and adjust both coarse and fine focus. The application points out that the magnification needs to be gradually changed or shows a broken slide if such a situation would occur in reality. There is also a test on the care and use of the microscope, calculation of magnification, and terminology. The *pedagogical suitability* was rated positively and the *interaction support* was rated neutral.

## Discussion

In our work, we identified the resources used and recommended by science teachers in five EU countries for teaching online science practicals. To recall our conceptualization, a *resource* is an objectively existing digital resource that teachers incorporate into their teaching in a (very) varied way, bringing technologies into their practice and creating technologically-enhanced activities. Among the reported resources, there were a few that were repeated across countries. These were PhET (Interactive Simulations for Science and Math, with many language mutations), Kahoot (in which teachers add content in their language), and three other resources mentioned only in Czechia and Slovakia where there is little, if any, language barrier. A large proportion of the reported resources were unique to countries. This situation highlights the need for an international design of circumstantial studies and raises two possible interpretations. Either the local resources are limited in their international impact by the language barrier (and the resulting low speed of dissemination into the global space), or the resources have some specificity to a given curriculum design or school tradition. This would be very interesting to find out, for example, in relation to cultural dimensions [81]. Teachers began to use approximately one third of the reported resources due to forced distance teaching. This shows that the respondents were open to including new resources and made them part of science education. They have undergone the two critical phases necessary for adopting teaching and learning based on computing devices formulated by [2] i) introduction and

gaining practice and ii) implementing the tools into lessons. It confirms that the data collection time was well chosen for such a type of study.

The vast majority of resources were used in multiple activities, even those that were not entirely expected on first impression of the content of the resource. This helped us to understand that the degree of variability with which teachers conceptualize the incorporation of a given resource, i.e., the nature of how they incorporate its use into their teaching and what kind of teaching activity they create from it, is enormous. We estimate that it exceeds the expectations of the creators of the resources–the creativity of the creators of resources is added to the creativity of the educators who push their work into sometimes unexpected positions. For this study, this creates a source of bias because we did not expect such diversity, and we took the resource as the primary entity of interest. Hence, it is not possible to determine whether, for example, added value refers to all types of use of a given resource, or whether teachers associated it with some particular specification of the activity. We interpret the data in accordance with the data collection concept with this potential bias in mind. Also, even though we asked about the modality [42], we did not use it in the analyses, because the teachers mostly reported both synchronous and asynchronous use of the resources. The resource mentioned most frequently was PhET, which also represented many of the most frequent characteristics; therefore, we use it as an example when discussing the findings of this study below.

To find out what types of activities were enhanced by using technologies, we analyzed their frequency. The activity mentioned most frequently was knowledge *revision*. This may imply the prevailing level of Bloom's taxonomy, although the activities most frequently mentioned, *revision* and *simulation*, also formed the most frequently mentioned activity couple, as in the example of PhET. And the revision using simulations suggests that in reality the teachers aimed at higher cognitive levels. The authors coded the level of Bloom's taxonomy (only as binary 'remembering'-'higher') because we believe that it helps better understand the role in teaching practicals. However, in this arrangement, the interpretation was difficult and sometimes even ambiguous because resources could be used in different ways.

The most important characteristics of the digital resources that led teachers to use them are described below. Again, the PhET resource had the characteristics important for the overall rating included in *pedagogical suitability*: ready to use without modifications to teach subject content and ICT, clear structure. Consequently, it was the best-rated resource within *pedagogical suitability*. This is in concordance with the literature, as PhET has been demonstrated to be an effective learning tool and it is gratifying that teachers across Europe are incorporating it into their (online science practicals) teaching [78, 79, 82].

As (social)interactions of different types are seen as one of the keystones of online education [24], we used them as a separate set of items and also added student-nature interaction, which is fundamental for science education. Our results show that teachers most recommended apps that promoted interaction between students, student and teacher, student and nature, and student and society. We hypothesize that teachers did not consider student + family interaction as important, or instead purposely encouraged interactions outside the family during lockdown, as it could be assumed that most of the students spent the majority of the lockdown time with their families. Interaction with peers is very important and it was challenging to maintain it during lockdowns [83]; therefore, it is good news that the teachers recommended the web resources that encourage student + student interaction. Student + teacher interaction, from which especially individual feedback is highlighted [84], was also supported by most of the web resources used. Students' interaction with nature is essential for both physical [85] and mental health [86] and should be one of the goals of science education. As such, it was considered important when choosing resources for online teaching. In the evaluation of

the case of PhET, the *interaction support* differed among teachers, as the web resource offers a wide range of different simulations and the specific task may vary.

As with most web resources, PhET is available for free and was used also before the forced online teaching brought about by COVID-19. Contrary to most web resources, PhET simulations represent many of the *added values*. Surprisingly, 36% of the recommended resources did not provide any *added value*, and these characteristics based on research and literature recommendations [24, 36, 37, 64, 66] did not influence the overall rating of the resources and were not in the regression model. Teachers evidently also used resources, which according to theory did not meet the requirements for online teaching of science practicals. The question is whether they were aware of this and used resources to add to other activities, or whether this was the result of forced online teaching that was not well thought out, lacked integration, and thus was unlikely to be effective [12, 13].

Based on the proposed model, the overall rating of the resources was influenced neither by Bloom's taxonomy nor by the *added values* derived from [24, 36, 64, 66]. Online teaching practice seems to be different from what theory assumes and demands if teachers still use the web resources as a part of effective online science teaching [12, 13]. On the other hand, it is possible that the usage of the web resources is not that well thought through and is a result of rather random choices and more integration is needed. The model shows that the *pedagogical support* and *interaction support* influence the overall rating. This seems to be straightforward, as the pedagogical support in addition to others describes how well the resource is prepared for direct usage in science education and the interactions are fundamental to education. To make resources ready for use in the classroom, their design is essential [23].

Although we expected a relationship between the properties of resource and activity, we had no theoretical basis for it. Based on the results of the classification model, we came to the following conclusions. Bloom's taxonomy was not significant in explaining the overall ranking, but was one of the most useful features in predicting activity type. This describes the reality of teaching in which teachers value both low and high Bloom levels according to what they need at the moment. Interestingly, using different organizational levels also differentiated the activities well. Explicit linking of multiple organizational levels is highly desirable because in science typically the (sub)molecular level determines all higher levels, and understanding all levels is essential for understanding scientific phenomena [66]. This is a phenomenon that deserves more attention in research. Other characteristics that are well applicable to differentiating activities were the worksheet, creativity, and school interaction. These characteristics are typical of science practicals. The most clearly delineated activities were games and virtual activities, which are the most technologically enhanced, but otherwise it is not clear why these are the ones. On the other hand, the most vaguely defined was knowledge revision, an activity for which (almost) a plethora of different resources was used. Similarly undefined were computer simulations as an activity highly variable both in its structure and in the possible resources. The wide variation among resources used for knowledge revision probably shows that resources with challenging tasks are lacking and replaced by proxy tools [23]. Additionally, teachers tend to use technology to assist traditional teaching and learning, not necessarily to implement reform-based practices [87].

## Limitations of the study

As in other similar studies, only teachers willing to participate in the survey shared their ideas with us. Not all science teachers in the countries were asked to participate, rather each institution used its own communication means to contact science teachers, which creates a bias. We only collected the web resources the science teachers would recommend; consequently, we do

not know which web resources they have tried and possibly rejected. Therefore, we cannot distinguish which ones they do not want to use and which ones they do not know. As mentioned above, a resource is taken as a base entity in our survey, and therefore, it is impossible to distinguish reported resource features from the specific digital element context.

## Conclusion

We built on previous work on the typology and description of the roles of digital elements in teaching and specified features that are essential for teachers' decision-making when integrating a given resource into teaching science practicals. By analyzing examples of good practice from five European countries, we found that teachers preferred web resources that provided knowledge revision and virtual simulations, that could be used without modification to teach content and ICT, and that promoted social and nature interactions. Pedagogical suitability (e.g., teaching content, ICT, clear structure) and support of interactions were the most influential factors in teachers' ratings. These factors determining the acceptance of the resource in science practicals did not differ considerably from the general trends either. From teachers' individual comments on the resources, it seems that the online form of teaching has even led to a blurring of the previously sharply defined boundaries between lesson and practicals, but more research would be needed to confirm this.

Surprisingly, characteristics considered essential by educational research experts, such as evaluation according to Bloom's taxonomy, added value (e.g., linking organizational levels, supporting independent solving, creativity, flexibility, allowing differentiation of difficulty levels, providing workshops) did not influence the overall rating of the web resources. However, these characteristics could be used to predict activity types linked to resources.

## Supporting information

**S1 Appendix. The table provides cleaned data used for analyses, that is, all the resources and their characteristics.**
(XLSX)

## Acknowledgments

We are grateful to the members of the My Home–My Science Lab project team and the teachers involved in the study.

## Author Contributions

**Conceptualization:** Vanda Janštová, Petr Novotný.

**Data curation:** Vanda Janštová.

**Formal analysis:** Petr Novotný.

**Funding acquisition:** Vanda Janštová.

**Methodology:** Vanda Janštová, Petr Novotný.

**Project administration:** Vanda Janštová.

**Software:** Petr Novotný.

**Writing – original draft:** Vanda Janštová, Petr Novotný, Irena Chlebounová, Fina Guitart, Ester Forne.

**Writing – review & editing:** Vanda Janštová, Petr Novotný, Montserrat Tortosa.

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
