## [Decision Letter · Decision Letter 0]

22 May 2023

PONE-D-23-02994Identifying key features of digital elements used during online science practicalsPLOS ONE

Dear Dr. Janštová,

Thank you for submitting your manuscript to PLOS ONE. After careful consideration, we feel that it has merit but does not fully meet PLOS ONE’s publication criteria as it currently stands. Therefore, we invite you to submit a revised version of the manuscript that addresses the points raised during the review process.

We look forward to receiving your revised manuscript.

Kind regards,

Tomislav Jagušt

Academic Editor

PLOS ONE

Journal Requirements:

"This work was supported by Erasmus+ project My Home - My Science Lab. We are grateful to the members of My Home - My Science Lab project and teachers involved in the study."

"All authors were funded by Erasmus+ project 'My Home - My Science Lab'.

Reviewers' comments:

Reviewer's Responses to Questions

**Comments to the Author**

1. Is the manuscript technically sound, and do the data support the conclusions?

Reviewer #1: Partly

Reviewer #2: Partly

2. Has the statistical analysis been performed appropriately and rigorously? 

Reviewer #1: No

Reviewer #2: I Don't Know

3. Have the authors made all data underlying the findings in their manuscript fully available?

Reviewer #1: Yes

Reviewer #2: Yes

4. Is the manuscript presented in an intelligible fashion and written in standard English?

Reviewer #1: No

Reviewer #2: Yes

5. Review Comments to the Author

Reviewer #1: This article presents interesting research on the topic of science practical online teaching during the COVID-19 lockdown. The research was carried out as a part of the Erasmus+ project "My Home – My Science Lab". In the research, the data was collected from science teachers in Slovakia, Czechia, Slovenia, France, and Spain regarding shared web resources they used and would recommend.

While the study's topic is interesting, the article's language is unclear and difficult to understand. It includes many errors, and I recommend that authors work with a writing coach or copyeditor to improve the flow and readability of the text. The paper should be written in an objective tone. Also, there are a lot of inconsistencies in the text, for example, the authors use both "eg." and "e.g." throughout the text, there is inconsistency in using small letters or caps (for example in Table 1), and inconsistency in the citation style used (for example line 507 with two different citation styles). Furthermore, for sentence separation the usage of ";" character instead comma is redundant, and the space character should be put after the comma, etc.

The authors are off to a good start, however, the study fails to address how the findings relate to previous research in this area. The authors should rewrite their Introduction to reference the related literature.

The sample size is not very high. Also, it seems as if the data collection process should have been better organized since, for example, the authors don't know which web resources teachers have tried and possibly rejected.

Although the research results are interesting, there is a lack of serious contributions and conclusions drawn from the research, for example, the nature of how teachers incorporate the use of resources into their teaching and what kind of "digital element" they created from it.

Some more specific comments:

Page 5, line 89: I would recommend describing "recommendations for conducting online teaching" in more detail.

Page 5, line 106: I would recommend describing "additional verbs typical for the digital environment" in more detail.

Page 6, line 113: Missing reference supporting "have been shown to be a challenging method even in face-to-face settings."

Page 10, line215: STEM is not a subject.

Page 23, line 406: I suggest reorganizing the paper in order to provide a description of applications used for online teaching at the beginning. Also, giving more insight into existing applications, not only just the one mentioned by the surveyed teachers.

Page 28, line 511: "i.e. the nature of how they incorporate its use into their teaching and what kind of ‘digital element’ they create from it, is enormous and we estimate that it exceeds the expectations of the creators of the resources" – this should be the base for the paper and would be interesting to see.

Page 31, line 580: "We only collected the web resources the science teachers would recommend; consequently we don't know which web resources they have tried and possibly rejected. Therefore, we cannot distinguish which ones they don't want to use and which ones they don't know." – this should have also been addressed during the data collection.

Reviewer #2: The paper presents research on identifying characteristics of digital elements used in science practical to identify which digital elements characteristics can serve as markers for their acceptance by teachers.

The authors should emphasize more the contribution of their research in relation to existing research, preferably at the end of the introduction.

The Introduction and Theoretical background section provide background information for readers, including motivation for the research. However, this part lacks a clear definition of the term "digital element" and a description of the relationship between the terms "digital element", "resource", and "learning activity".

The authors should distinguish the terms (concepts) "resource" and "learning activity". It seems that these terms are sometimes considered synonymous, which makes it difficult to follow the manuscript and understand the analysis performed and its results. In the introduction, the authors give the following examples of digital elements: videos, visual materials, and domain-specific simulations, but later they also mention investigations, experiments, games, modeling and construction, etc. as types of digital elements. For example, in the learning activity "knowledge revision", students can be referred to different web resources (e.g. a specific video), so it is necessary to pay attention to the terms used.

Also, before describing the importance of "science practicals", it would be desirable for the authors to describe in more detail what they mean by this term and provide some examples.

The description of the methodology does not specify the research questions. Formulating the research questions using appropriate terms would allow for better structuring of the sections in which the research findings and discussion are presented.

Regarding the instrument, it is necessary to better justify the selection of the types of digital elements used to describe digital elements and to present all the statements used to classify them according to other criteria (e.g., pedagogical suitability).

In the abstract, the sentence that starts with "We identified their key..." is unclear.

6. PLOS authors have the option to publish the peer review history of their article (what does this mean?). If published, this will include your full peer review and any attached files.

Reviewer #1: No

Reviewer #2: No

---

## [Author Response · Author response to Decision Letter 0]

4 Jul 2023

Thank you for the summary and the detailed comments from the reviewers.

We have edited the manuscript according to the comments, in particular we have unified the terminology, rewritten the introduction and added one analysis that allowed us to formulate a further conclusion. Detailed responses to each comment are provided below.

—

Comments to the Author

1. Is the manuscript technically sound, and do the data support the conclusions?

Reviewer #1: Partly

Reviewer #2: Partly

2. Has the statistical analysis been performed appropriately and rigorously?

Reviewer #1: No. 

Reviewer #2: I Don't Know

3. Have the authors made all data underlying the findings in their manuscript fully available?

Reviewer #1: Yes

Reviewer #2: Yes

4. Is the manuscript presented in an intelligible fashion and written in standard English?

Reviewer #1: No

Reviewer #2: Yes

5. Review Comments to the Author

Reviewer #1: This article presents interesting research on the topic of science practical online teaching during the COVID-19 lockdown. The research was carried out as a part of the Erasmus+ project "My Home – My Science Lab". In the research, the data was collected from science teachers in Slovakia, Czechia, Slovenia, France, and Spain regarding shared web resources they used and would recommend.

While the study's topic is interesting, the article's language is unclear and difficult to understand. 

It includes many errors, and I recommend that authors work with a writing coach or copyeditor to improve the flow and readability of the text. The paper should be written in an objective tone. Also, there are a lot of inconsistencies in the text, for example, the authors use both "eg." and "e.g." throughout the text, there is inconsistency in using small letters or caps (for example in Table 1), and inconsistency in the citation style used (for example line 507 with two different citation styles). Furthermore, for sentence separation the usage of ";" character instead comma is redundant, and the space character should be put after the comma, etc.

Thank you for the comment, the whole text was checked and corrected, the citation style was unified.

2. The authors are off to a good start, however, the study fails to address how the findings relate to previous research in this area. The authors should rewrite their Introduction to reference the related literature.

The introduction was re-written to cover a wider context.

3. The sample size is not very high. Also, it seems as if the data collection process should have been better organized since, for example, the authors don't know which web resources teachers have tried and possibly rejected. 

As we aimed at describing the digital resources which teachers used, we omitted the idea of asking for the resources they had tried and rejected. One can hardly assume that teachers, when searching for resources, will remember which resources they have not accepted, let alone attach relevant assessments to them retrospectively. . Based on the literature and also our results, we conclude (at least some) teachers were actively searching for new web resources during forced distant teaching (32 out of 89 web resources from our data set were discovered by the teachers during forced distant teaching). Still, there is a nearly endless supply of resources that is being broadened every day.

4. Although the research results are interesting, there is a lack of serious contributions and conclusions drawn from the research, for example, the nature of how teachers incorporate the use of resources into their teaching and what kind of "digital element" they created from it. 

Thank you for your comment. These are interesting questions that mostly go beyond the scope we are working in. But based on it, we additionally analyzed the relationship between features of the digital resource and activity type (in which the resource was used by the teachers) and added it to the manuscript.

Some more specific comments:

5. Page 5, line 89: I would recommend describing "recommendations for conducting online teaching" in more detail. More detailed description added.

Page 5, line 106: I would recommend describing "additional verbs typical for the digital environment" in more detail. Examples added.

Page 6, line 113: Missing reference supporting "have been shown to be a challenging method even in face-to-face settings." Reference added.

Page 10, line215: STEM is not a subject. Corrected.

6. Page 23, line 406: I suggest reorganizing the paper in order to provide a description of applications used for online teaching at the beginning. Also, giving more insight into existing applications, not only just the one mentioned by the surveyed teachers.

A selection of the applications used was moved at the beginning of Results. We did not add a list or description of other existing applications as there is a huge number of them and we did not want to choose them subjectively based on our knowledge.

7. Page 28, line 511: "i.e. the nature of how they incorporate its use into their teaching and what kind of ‘digital element’ they create from it, is enormous and we estimate that it exceeds the expectations of the creators of the resources" – this should be the base for the paper and would be interesting to see. 

We agree this would be interesting and we see it as a possible next step of research. However, to have the base for it, we wanted to describe the web resources in this manuscript. See comment 4

8. Page 31, line 580: "We only collected the web resources the science teachers would recommend; consequently we don't know which web resources they have tried and possibly rejected. Therefore, we cannot distinguish which ones they don't want to use and which ones they don't know." – this should have also been addressed during the data collection. 

Added to “Sample and sampling”.

Reviewer #2: The paper presents research on identifying characteristics of digital elements used in science practical to identify which digital elements characteristics can serve as markers for their acceptance by teachers.

9. The authors should emphasize more the contribution of their research in relation to existing research, preferably at the end of the introduction.

Different characteristics are important for different perspectives that teachers use when preparing their lessons. See comment 4

10. The Introduction and Theoretical background section provide background information for readers, including motivation for the research. However, this part lacks a clear definition of the term "digital element" and a description of the relationship between the terms "digital element", "resource", and "learning activity".

Thank you, indeed. The terminology was clarified. We consider the variability of incorporation to be necessary to explore through an observational approach (activity structures) and the typology of characteristics of digital resources may be a useful tool for this. Based on reviewers feedback, we have modified the terminology - although the term digital element seemed to be more apt for today's reality (when almost all ICT technologies are “transparent”), we have reverted to the term (learning) activity for clarity. The term online resource, digital resource, and resource is used in the meaning of a particular web page, and activity is one of the characteristics of the resources.

 - see also comment 4

11. The authors should distinguish the terms (concepts) "resource" and "learning activity". It seems that these terms are sometimes considered synonymous, which makes it difficult to follow the manuscript and understand the analysis performed and its results. In the introduction, the authors give the following examples of digital elements: videos, visual materials, and domain-specific simulations, but later they also mention investigations, experiments, games, modeling and construction, etc. as types of digital elements. For example, in the learning activity "knowledge revision", students can be referred to different web resources (e.g. a specific video), so it is necessary to pay attention to the terms used.

Thank you, agree, see comments 4+10

12. Also, before describing the importance of "science practicals", it would be desirable for the authors to describe in more detail what they mean by this term and provide some examples. 

The term science practicals has been defined and examples given.

13.The description of the methodology does not specify the research questions. Formulating the research questions using appropriate terms would allow for better structuring of the sections in which the research findings and discussion are presented.

Thank you, research questions were formulated and used for further structuring the text.

14. Regarding the instrument, it is necessary to better justify the selection of the types of digital elements used to describe digital elements and to present all the statements used to classify them according to other criteria (e.g., pedagogical suitability).

The selected characteristics of digital resources are described in theoretical background in more detail and the statements were added to methods - instrument.

15. In the abstract, the sentence that starts with "We identified their key..." is unclear.

The sentence was reformulated.

---

## [Decision Letter · Decision Letter 1]

24 Aug 2023

PONE-D-23-02994R1Identifying key features of digital resources used during online science practicalsPLOS ONE

Dear Dr. Janštová,

Thank you for submitting your manuscript to PLOS ONE. After careful consideration, we feel that it has merit but does not fully meet PLOS ONE’s publication criteria as it currently stands. Therefore, we invite you to submit a revised version of the manuscript that addresses the points raised during the review process.

We look forward to receiving your revised manuscript.

Kind regards,

Tomislav Jagušt

Academic Editor

PLOS ONE

Journal Requirements:

Reviewers' comments:

Reviewer's Responses to Questions

**Comments to the Author**

1. If the authors have adequately addressed your comments raised in a previous round of review and you feel that this manuscript is now acceptable for publication, you may indicate that here to bypass the “Comments to the Author” section, enter your conflict of interest statement in the “Confidential to Editor” section, and submit your "Accept" recommendation.

Reviewer #1: All comments have been addressed

Reviewer #2: All comments have been addressed

2. Is the manuscript technically sound, and do the data support the conclusions?

Reviewer #1: Yes

Reviewer #2: Yes

3. Has the statistical analysis been performed appropriately and rigorously? 

Reviewer #1: Yes

Reviewer #2: Yes

4. Have the authors made all data underlying the findings in their manuscript fully available?

Reviewer #1: Yes

Reviewer #2: Yes

5. Is the manuscript presented in an intelligible fashion and written in standard English?

Reviewer #1: No

Reviewer #2: Yes

6. Review Comments to the Author

Reviewer #1: The authors have improved their paper, yet the majority of the paper shoud be rewritten for clarity. Please consult a native English speaker to rewrite sentences.

Introduction should be rewritten. Introduction as whole is rather poorly structured and unclear. Please consult a native English speaker to rewrite sentences. For example:

- "obstacles that were often considered objective limiting factors to the introduction of technology into the classroom" - what obstacles? please reference or write examples

- The frequency of their use and the types of digital resources (tools that are part of computing

environments to support education [1] that teachers used, the method and media, but even the

modality varied depending on the personality of the teacher, the technological equipment of

the school, and also the school philosophy. Our focus is on the practice-oriented components

of science teaching, by which we mean hands-on exercises, lab work (referred to as science

practicals in this study) which, at least in our experience, have been implemented online

rather infrequently. -this is very unclear, please rewrite

“Therefore, when preparing designing online learning environments, socio-emotional processes have not been the main focus [7].” – very unclear

Furthermore, why is Information and Communication Technology written in caps? I would suggest small letters.

Reviewer #2: The authors have improved the quality of the manuscript in accordance with my comments and concerns so I propose to accept the current version of their manuscript for publication.

7. PLOS authors have the option to publish the peer review history of their article (what does this mean?). If published, this will include your full peer review and any attached files.

Reviewer #1: No

Reviewer #2: No

---

## [Author Response · Author response to Decision Letter 1]

21 Sep 2023

Dear editor and reviewers,

Thank you for the summary and the detailed comments. 

We have reviewed our reference list, ensuring that it is accurate and up-to-date and contains no retracted references. 

We revised the introduction and the entire manuscript to improve clarity. To ensure linguistic precision, we sought assistance from a native English speaker. Your feedback on specific cases was appreciated, and we addressed these concerns in our revised submission.

On behalf the author team,

Vanda Janštová

---

## [Editor Report · Decision Letter 2]

10 Oct 2023

Identifying key features of digital resources used during online science practicals

PONE-D-23-02994R2

Dear Dr. Janštová,

We’re pleased to inform you that your manuscript has been judged scientifically suitable for publication and will be formally accepted for publication once it meets all outstanding technical requirements.

Kind regards,

Tomislav Jagušt

Academic Editor

PLOS ONE
---

## [Editor Report · Acceptance letter]

13 Oct 2023

PONE-D-23-02994R2 

Identifying key features of digital resources used during online science practicals 

Dear Dr. Janštová:

I'm pleased to inform you that your manuscript has been deemed suitable for publication in PLOS ONE. Congratulations! Your manuscript is now with our production department. 

Kind regards, 

on behalf of

Dr. Tomislav Jagušt 

Academic Editor

PLOS ONE